# Technology of Automatic Evaluation of Dairy Herd Fatness

**Sergey S. Yurochka** [1], **Igor M. Dovlatov** [1,*], **Dmitriy Y. Pavkin** [1], **Vladimir A. Panchenko** [2],
**Aleksandr A. Smirnov** [1], **Yuri A. Proshkin** [1] and **Igor Yudaev** [3]

1   Federal State Budgetary Scientific Institution, Federal Scientific Agroengineering Center VIM (FSAC VIM), 1st Institutsky Proezd 5, 109428 Moscow, Russia; yurochkasr@gmail.com (S.S.Y.); dimqaqa@mail.ru (D.Y.P.); alexander8484@inbox.ru (A.A.S.); ledsk@bk.ru (Y.A.P.)

2   Department of Theoretical and Applied Mechanics, Russian University of Transport, 127994 Moscow, Russia; pancheska@mail.ru

3   Energy Department, Kuban State Agrarian University, 350044 Krasnodar, Russia; etsh1965@mail.ru

\*   Correspondence: dovlatovim@mail.ru

**Abstract:** The global recent development trend in dairy farming emphasizes the automation and robotization of milk production. The rapid development rate of dairy farming requires new technologies to increase the economic efficiency and improve production. The research goal was to increase the milk production efficiency by introducing the technology to automatically assess the fatness of a dairy herd in 0.25-point step on a 5-point scale. Experimental data were collected on the 3D ToF camera O3D 303 installed in a walk-through machine on robotic free-stall farms in the period from August 2020 to November 2022. The authors collected data on 182 animals and processed 546 images. All animals were between 450 and 700 kg in weight. Based on the regression analysis, they developed software to find and identify the main five regions of interest: the spinous processes of the lumbar spine and back; the transverse processes of the lumbar spine and the gluteal fossa area; the malar and sciatic tuberosities; the tail base; and the vulva and anus region. The adequacy of the proposed technology was verified by means of a parallel expert survey. The developed technology was tested on 3 farms with a total of 1810 cows and is helpful for the non-contact evaluation of the fatness of a dairy herd within the herd's life cycle. The developed method can be used to evaluate the tail base area with 100% accuracy. The hungry hole can be determined with a 98.9% probability; the vulva and anus area—with a 95.10% probability. Protruding vertebrae—namely, spinous processes and transverse processes—were evaluated with a 52.20% and 51.10% probability. The system's overall accuracy was assessed as 93.4%, which was a positive result. Animals in the condition of 2.5 to 3.5 at 5–6 months were considered healthy. The developed system makes it possible to divide the animals into three groups, confirming their physiological status: normal range body condition, exhaustion, and obesity. By means of a correlation dependence equal to R = 0.849 (Pearson method), the authors revealed that animals of the same breed and in the same lactation range have a linear dependence of weight-to-fatness score. They have developed an algorithm for automated assessment of the fatness of animals with further staging of their physiological state. The economic effect of implementing the proposed system has been demonstrated. The effect of increasing the production efficiency of a dairy farm by introducing the technology of automatic evaluation of the fatness of a dairy herd with a 0.25-point step on a 5-point scale had been achieved. The overall accuracy of the system was estimated at 93.4%.

**Keywords:** dairy cows; body condition score; 3D TOF sensor; non-contact evaluation; recognize area of interest

## 1. Introduction

Over the last few decades, the global trend in dairy farming has been to automatize and robotize milking processes on commercial farms [1,2]. The common average production period of dairy animals is 3.5 lactations [3]. Due to the rapid development of dairy farming,

new technologies are increasingly required to achieve a higher economic efficiency and achieve an improved production [4–6]. On the one hand, intensive production results in an increased milk yield of a cow; on the other hand, intensive production leads to the rapid deterioration of dairy cows—i.e., a reduction in the number of lactations [7]. The reduction of the production life of dairy animals also depends on the premature culling of animals that have a high or low body condition score. Lack of a normal body condition score during lactation is primarily due to dietary deficiencies [7]. Another negative consequence is culling of animals due to poor body condition because an increased body condition score reduces fertility and thereby extends the service period.

A body condition score (BCS) evaluation is important in technological milk production. First and foremost, the BCS score is used to place animals within productivity groups and determine their status. In Russian dairy farm conditions, veterinarians and livestock breeding technicians rotate animals into production groups once a month, provided the milk production technology is well established. The body condition score helps make a decision individually for each cow, based on her current physiological condition, rather than simply on accepted technological norms. In intensive milk production, dairy cows are divided into 5 main groups: group 1, the step-ladder milk yield increasing group, includes new cows from 6 to 100 days after calving, and also cows with a daily milk yield of more than 24 kg per head per day. The total productivity of this group of animals should not be lower than 6000 kg per head per year. The main objectives of the group are: quality feeding with full-fat mixes and good care to achieve the peak milk production by day 40–50; elimination of post-calving complications to inseminate the animals on day 65. During this period, the animals give up to 65–70% of their milk volume during the lactation period. High-yielding cows are transferred to group 1 and should be in group 2, but they need increased nutrition according to milk yield and body condition score. The typical fatness score for group 1 is 3.5 to 3.25 from day 6 to day 30, and 3 to 2.75 from day 31 to day 100. The normal decrease of the body condition score of cows in group 1 is due to an intensive milk production, which requires a large amount of energy. The energy expended cannot be fully compensated by the energy gained from feeding. Therefore, it results in a natural decrease of the body condition score. Maria Ledinek et al., in a study [8], showed that during the calving period, the body condition score decreases, and body fat reserves provide for an increased milk production. By 40–65 days after lactation, animals should be milked as often as possible, and the body condition score should not be reduced by more than 0.5. During this period, the cow consumes up to 12 kg of high energy feed. Insemination takes place when the animal is at peak production and consumes the highest amount of feed and the fatness score is within 3 points. At the same time, the animal's body condition score may not deviate by more than ±0.25 points.

Group 2—milking cows from 101 to 305 days after calving with 24 to 16 kg of milk per head per day. The main objective for the animals in this group is to ensure that the milk yield does not fall by more than 9% per month, and to increase the body condition to 3–3.5 fatness points.

Group 3—milking cows from 101 to 305 days after calving with a milk yield below 16 kg/head/day. The main task for this group is to prevent diseases, correct body weight to a fatness of 3.5–3.75 points and prepare the animals for drying off.

When animals are in the second and third physiological group, from 101 to 305 days after calving, it is necessary to monitor their condition. A cow should have 3.5–3.75 by the start of the dry period. If it is under-conditioned, it should be kept in the first or second group and its milk production should be ignored. Otherwise, under-conditioning can lead to complications during parturition or at the beginning of the next lactation [9]. Overconditioning of a cow above 3.75 will result in an increase in fetal weight. As the cow's weight increases, so does the calf's weight. An increased calf weight at calving causes birth complications and injuries that are equally detrimental to the cow and calf.

Group 4—the first 45 days from day 306 after drying off. During this period, no adjustment is made to the animal's body condition score. It is assumed that the animal already has a body condition score of 3.5–3.75.

Group 5 is the maternity group. The animals are kept 15 days before calving and 5 days after calving. During this period, no adjustment is made to the body condition score of the animals.

The fatness assessment of dairy cows is not only a valuable indicator for evaluating the quality of feeding and the response of the animal to feed, but is also an indirect indicator of its reproductive function. Dairy performance correlates with feed intake. An increase in milk production is associated with a decrease in fertility. During peak lactation, cows require 3.5 times more protein and energy for milk synthesis than protein and energy for life support, as lactation and calf feeding have a higher biological priority than body weight gain and fecundity. At peak milk production, quit estrus and overcalving are the most significant problems. A negative energy balance, which is also affected by decreased body condition dynamics, results in a delayed onset of first heat and ovulation after calving in underfed cows, reduced probability of conception after first insemination, negative effects on follicle growth, corpus luteum function, oocyte quality, impaired intrauterine development, and embryo survival and growth [10].

Cows with a body condition at day 60 of 3.25–2.75 have a 67% chance of conception, and those with a body condition below 2.75 have a 44% chance [11].

Mohamed A.B. Mandour, in a study [12], found that high fatness in first-year heifers increases the risk of ketosis to 3.71%, which is twice as high as in adult cows. The study mentions that cows with a high body condition score consume less feed than cows with a normal body condition score and have a high negative energy balance due to a higher concentration of fatty acids in the plasma, which is associated with an increased risk of ketosis.

Thinawanga Joseph Mugwabana et al., in a study [13], found no relationship between the fatness of cows and the calving rate.

Wynnton C. Meteer [14] found in their study that animals given 70% of the required feed energy had more embryos at the next insemination and a higher probability of insemination than animals that received an energy excess of 130% of the norm. Changing the level of feeding in animals in groups 4 and 5 (middle and late stage before calving) did not significantly affect the amount of pregnancy hormones excreted in the blood.

These studies confirm the above information that the main management of feeding, control, and changing the body condition score of cows should be done during lactation, in animals in groups 1 and 2, to increase the probability of reproductive success in the next insemination of animals.

Poczta W. et al., in their study [15], established a relationship between cow fatness and the likelihood of subclinical ketosis, where cows with a fatness score $\geq 3.25$ were more susceptible to the disease than lean cows with a fatness score $\leq 3$.

Vanholder T. et al., in a study [16], found a relationship between the body condition score of dairy cows and weight loss within 30–40 days after calving. Of the 47 cows studied, 37 cows lost $\geq 0.75$ BCS points at 14 days post calving, and 10 cows lost $\leq 0.75$ BCS points. Weight loss is associated with a negative energy balance in the cow after calving and subsequent mobilization of body reserves for recovery.

In [17], the authors found a correlation between the propensity for metritis and the BCS of cows $\leq 3$. In [18], the authors evaluated the relationship between BCS points during the transition period and the development of disease and changes in milk yield. A total of 232 cows were assessed and the fatness was scored from 1 to 5 in a 0.25 step. After a blood test, a conclusion on the health of the animals was made. Changes in the body condition of dairy cows using the BCS scale were measured at 21 days before calving and 21 days after calving. The percentage of cows that increased BSC (fatness) during this period was 28%, lost BCS—22%, and retained BCS—50%. Additionally, 18% of the cows that lost BCS during this period had health problems compared to the cows that retained the BCS points.

Furthermore, 28% of the cows that had an increased BCS were less likely to have subclinical ketosis.

The results confirmed that developing ketosis can be detected in an automatic, non-contact method. An alternative way of detecting ketosis is presented in studies [19–21], where blood tests were required to detect disease. On large farms with more than 200 milking herds, the continuous active assessment of animal health by blood testing is not possible, due to the lack of specialists, the time-consuming process, and the need for laboratory equipment. The BCS can be evaluated both manually and automatically.

Studies [18–22] describe the manual method of BCS evaluation. Study [23] gives a detailed review of automatic systems for automatic BCS evaluation. Study [24] describes the development of an automatic BCS evaluation system using a deep learning neural network algorithm using a convolutional neural network. The researchers achieved a recognition accuracy of 94% at a step of 0.5, and 78% at a step of 0.25. In [25], the authors used a convolutional neural network (CNN) to evaluate BCS. The accuracy of the system was assessed using the Kappa index and was within a moderate range (values between 0.41 and 0.60). In [26], the authors also used a convolutional neural network (CNN) to evaluate BCS. The accuracy of the results obtained in the study was 78%, indicating a successful real-time classification. In [27], the authors used the point cloud method to evaluate BCS. Experiments show that the proposed BCS evaluation model achieved an accuracy of 49, 80, and 96% within a deviation of 0, 0.25, and 0.50 points, respectively.

In [28], a dynamic background model (Gaussian Mixture Model, GMM) was used to distinguish the cow from the background. Subsequent Image Processing Algorithms have made it possible to automatically obtain reliable images, to find areas of interest, and to extract image elements without any manual intervention. With 5-fold cross checking, the model has achieved an average accuracy of 56% with a 0.125-point variance, 76% with a 0.25-point variance, and 94% with a 0.5-point variance.

Having studied the modern experience of the world community on the automation of BCS evaluation, our team had set a goal and fully fulfilled the tasks on the development of technology of an automatic system of BCS evaluation. The aim of the research was to improve the production efficiency of dairy farms by implementing the technology of an automatic BCS evaluation of a dairy herd with a 0.25-point step on a 5-point scale.

The main approach we used in developing the technology was to minimize the use of neural network algorithms to find areas of interest. This decision was based on the experience of the team [29,30]. Training neural networks was a labor-intensive and time-consuming process. For example, a trained neural network for standardized breeds of EU countries—Holstein, Brown Latvian, Swiss, etc.—will give a big error during a BCS evaluation of Black-Motley Holstein, Kalmyk breeds, etc. To minimize the error, it was necessary to retrain the neural network. The method we proposed is based on the study of standardized breeds of EU countries to adjust the model and to carry out a further BCS evaluation, avoiding the training of the neural network algorithm on each farm.

Thus, the research resulted in the development of a universal automatic system capable of estimating the BCS of an animal with high accuracy (more than 90%) at a step of 0.25. The developed system was intended for the implementation in automated and robotic free-stall farms. The system was designed to evaluate the BCS (fatness) of animals and to provide recommendations for a wider range of functions to be carried out by a specialist.

## 2. Materials and Methods

### 2.1. Farm, Field Data Collection

In earlier studies, we had already achieved a result of algorithmic evaluation, where the system evaluated a fatness score between 2 and 4 with a 10% error, while scores 1 and 5 were evaluated with a 25% error (the results of the automatic evaluation were checked against the results of an expert panel) [29,30]. In the study we conducted, the unsatisfactory result that required further research was on the evaluation of the boundary body condition scores 1 and 5. The difficulty lies in the fact that for the algorithm, cows with a body

condition score of 1 and 2 and a score of 4 and 5 are similar. Therefore, in this study, we focused our field data collection on animals with a body condition score of 1 to 3 and 4 to 5.

We selected 3 commercial farms with a total of 182 animals. On the selected farms, all cows have similar traits, the animals are emaciated and of poor performance, and part of the herd features are overweight. Data on animals were collected in the Moscow and Yaroslavl regions.

Field data were collected between August 2020 and November 2022:

- The first farm has a herd of 570 forage cows, which is located in the Yaroslavl region (Central Russia). On this farm, 118 animals were selected at 5–6 months of lactation (from 151 to 180 days of lactation), with a body condition score of 1–4 points. The average annual milk yield per cow per day is 16.8, with an average fat percentage of 3.7%. The animals are of the black-motley breed. The average body condition score of the experimental animals was 2.75. The average weight of the tested animals is 467 kg.
- The second farm has a forage-fed herd of 50 heads, located in the Moscow region (Central Russia). On this farm, we selected 18 animals of 5–6 months of lactation (from 151 to 180 days of lactation) with a body condition score from 2 to 5. The average body condition score of the sample animals was 3.75. The average annual milk yield per cow per day is 15 kg/milk. The average fat content in the milk is 5.2%. The animal breed is the Holsteinized black-motley breed. The high fat content of milk of the black-motley breed can be explained by the fact that additional local selection work was done on this farm to increase the fat content of milk. The milk obtained from the animals is used for cheese production. The average weight of the tested animals is 583 kg.
- the third farm has a herd of 1200 forage-fed animals and is located in the Moscow region (Central Russia); 46 animals of 5–6 months of lactation (from 151 to 180 days of lactation) and with a body condition score of 3–5 were selected on this farm. Experimental animals had an average body condition score of 3.5. The breed of animals were Holsteinized black and mixed breeds. A total of 36 animals were selected, 5–6 months of lactation, which predominantly had a borderline body condition score. The average annual milk yield per cow per day was 28 kg/milk, and the average fat content of the milk was 3.7%. The average weight of the studied animals was 571 kg.

All animals had two milkings per day. Expert panels were formed to assess the body condition score by data collection site. The panel consisted of at least two independent veterinarians and two trained specialists. The average body condition score obtained from all experts was the benchmark value.

In addition, a weighing platform was used as an implement to further increase the accuracy of the body condition score by comparing the values obtained.

The live weight of animals was collected, in particular, by the Klüver–Strauch method [31], and the remaining animals were weighed on the platform. A disadvantage of the Klüver–Strauch method is an error in measuring the live weight of up to 10%. In the experiment, a discount of 1% of the actual weight was taken into account for bulk (mud adhered to the animal), and a discount of 3% for the contents of the gastrointestinal tract, when animals were weighed on the platform. Calculation of the live weight, including the discounts made, was automatic.

Based on the Pearson's correlation coefficient, we obtained proof of the representativeness of the animal samples and the relationship between the body condition score and live weight of the animals.

### 2.2. Equipment and System

Images of the cows' backs were collected on a 3D commercial camera, the O3D 303 3D ToF camera. The system is powered by 220 V, the power supply converts to 24 V to ensure the 3D sensor is operational. The sensor is mounted at a height of +2.2 m above the floor level (Figure 1).

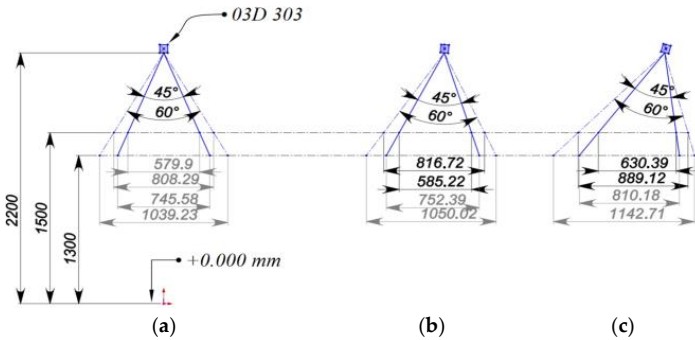

**Figure 1.** Demonstration of how the distance of the scanned surface varies with the inclination angle of the sensor. (**a**) Position 1 of the optical module 03D "looks" vertically down relative to the back of the cow; (**b**) position 2 of the optical module 03D "looks" at an angle of 5° from the vertical axis; (**c**) position 3 of the optical module 03D "looks" at a 10° angle from the vertical axis.

The height and inclination angle of the three-dimensional sensor are based on four parameters: cow height, cow length, minimum working distance between the camera and the object, and the camera's allowable error. The height of the animals under study ranged between 1300 mm and 1500 mm, the minimum working distance of the camera between the surface and the object under study was 300 mm. The signal from the identification antenna of the animal's RFID tag triggered the three-dimensional image production.

The inclination angle of the sensor taking into account the given 4 parameters is chosen to be 5 degrees, as it can cover a sufficient area of the animal's back under analysis, while keeping the pixel spacing to 0.006 m as the point of interest moves away from the 3D camera. The distance of 0.006 m between pixels is the set distance on which the least squares method is based when forming clusters of points related to areas of interest.

For the correct calculation of the required parameters between the camera and the object under study (coordinates of the received Z-axis pixels), we performed angle normalization (because the tilt angle of the 3D camera relative to the cow's back was introduced), presented in the expression using the R matrix:

where X, Y, Z—the areas of interest point coordinates, and J—the required distance between the interest points areas.

The total dataset contained 546 images from 182 animals with body condition scores from 1 to 5 with a step of 0.25 points (17 classes) (Table 1). Based on the earlier studies, it is sufficient for this camera to take three pictures of each cow, then the images are combined and the system starts determining the fatness.

**Table 1.** Number of images and proportion of cows for each body condition score.

| | Body Score Condition | | | | | | | | | | | | | | | | |
|---|---|---|---|---|---|---|---|---|---|---|---|---|---|---|---|---|---|
| № | 5 | 4.75 | 4.5 | 4.25 | 4 | 3.75 | 3.5 | 3.25 | 3 | 2.75 | 2.5 | 2.25 | 2 | 1.75 | 1.5 | 1.25 | 1 |
| * | 5 | 14 | 4 | 12 | 7 | 5 | 8 | 14 | 18 | 25 | 14 | 11 | 11 | 6 | 12 | 9 | 7 |
| ** | 15 | 12 | 12 | 36 | 21 | 15 | 24 | 42 | 52 | 75 | 42 | 33 | 33 | 18 | 36 | 27 | 21 |

\*—the number of animals; \*\*—the number of images.

From the data collected, we can see that the predominant body condition scores are 4.75; 4.25; 3.25; 3; 2.5; 2.25; 2; 1.5. The distribution of animals by body condition score was made by the expert panel, whose opinion is considered to be the benchmark (Figure 2).

The 3D camera is able to calculate and output Point Cloud as a multidimensional array I × J × K, where I and J are camera resolution, e.g., 352 × 264, K is X, Y, Z coordinates. Output of received data is in "dat" and ".h5" formats. The recording speed of the video images is 5 fps. Due to this feature, we obtained 3 to 5 images of each cow in the initial image. The images were collected according to the scheme shown in Figure 1. The camera

error stated in the manufacturer's specifications is ±0.01 m for each meter between the lens and the object. Therefore, assuming that the working distance between the cow's rump (1.5 m) and the 3D camera lens (2.2 m) is 0.4 m, the error amounted to ≤0.01 m. The optical module was installed at an angle of 5°.

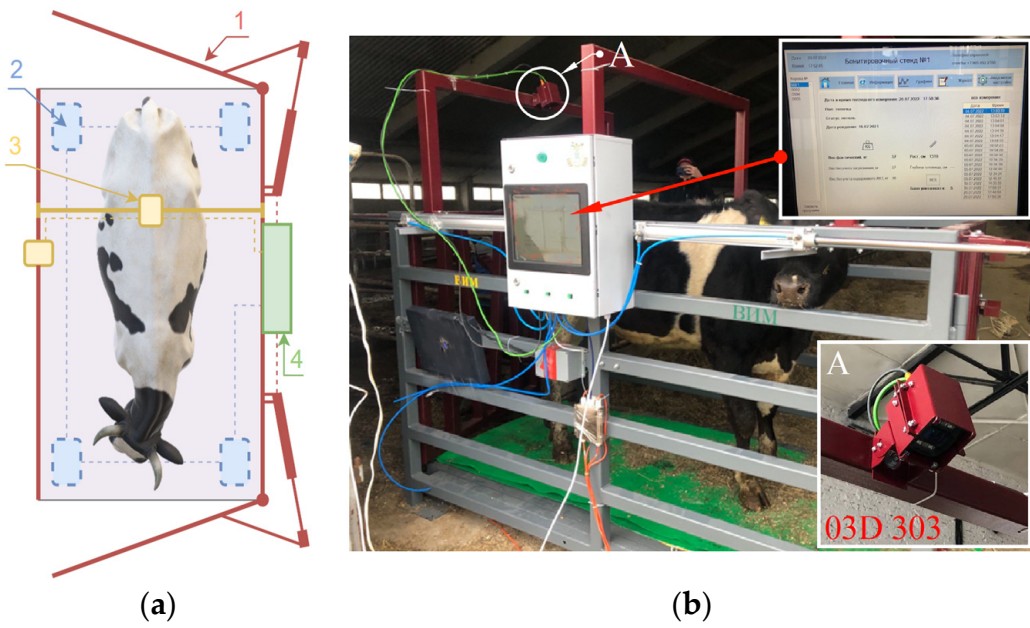

**Figure 2.** Developed installation used for field data collecting. (**a**) Scheme of the developed installation to determine the body condition score, height, and weight of dairy cows up to 1200 kg: 1—automatic gates; 2—weighing module; 3—03D 303 three-dimensional camera; 4—a single control unit; (**b**) three-dimensional camera for the body condition score evaluation—03D 303 and software.

*2.3. Assessment of the Body Condition Score and Analysis of the Results*

We used our previously developed software [32] to process the obtained three-dimensional maps and determine the body condition score and standard tools; excel for primary data processing and formatting was used to process the study results.

The results were obtained automatically and those of an expert evaluation were compared manually. The expert evaluation of the body condition score was a benchmark value.

In terms of searching and determining the main areas of interest, the developed software was based on the application of the least squares method (regression analysis) to find the areas of interest.

As the camera was mounted on top of the animal and the points of the cow's back were presented to the data analysis, the points of greatest interest were those near the contour and describing its perimeter. Using the spine of a cow as an example, we can consider the basic expressions to identify it. The algorithm developed is based on the ordinary method of least squares (LS).

The entire surface of a cow's back is represented by an array of points without regard to depth, after which the regression tool is applied. We represent the whole surface as a set of points:

$$(y_1, x_1), (y_2, x_2), \ldots (y_n, x_n) \tag{1}$$

We can apply the method of least squares to minimize the sum of squares of RSS RRS deviations:

$$\text{RSS} = \sum_i (y_i - (a + bx_i))^2 \tag{2}$$

To find fixed points for RSS, the following expressions are used:

$$\begin{cases} \dfrac{\partial RSS}{\partial a} = \sum_i 2(y_i - a - bx_i) = 0 \\ \dfrac{\partial RSS}{\partial b} = \sum_i 2(y_i - a - bx_i) = 0 \end{cases} ; \tag{3}$$

$$\begin{cases} \sum_i y_i - na - b\sum_i x_i = 0 \\ \sum_i x_i y_i - a\sum_i x_i - b\sum_i x_i^2 = 0 \end{cases} \tag{4}$$

$$\begin{cases} \overline{y} - a - b\overline{x} = 0 \\ \overline{xy} - a\overline{x} - b\overline{x^2} = 0 \end{cases} \tag{5}$$

$$\begin{cases} a = \overline{y} - b\overline{x} \\ \overline{xy} - (\overline{y} - b\overline{x})\overline{x} - b\overline{x^2} = 0 \end{cases} \tag{6}$$

$$\begin{cases} a = \overline{y} - b\overline{x} \\ \overline{xy} - \overline{x}\,\overline{y} + b\left[(\overline{x})^2 - \overline{x^2}\right] = 0 \end{cases} \tag{7}$$

Thus, the regression and refinement of the ridge line to the point cloud produces the result shown in Figure 3.

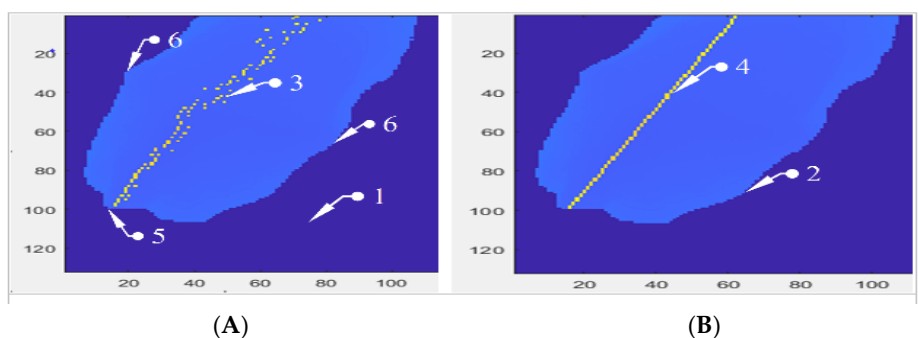

(**A**)                (**B**)

**Figure 3.** Defining the spinal column axis with extracting the area of interest. 1—Filtered area; 2—cow's contour; 3—unspecified animal's spinal column; 4—specified animal's spinal column; 5—tail head; 6—hips. (**A**) initial ridge line plotting by linear regression. (**B**) the ridge construction as a set of points on each longitudinal axis.

Figure 3 showed the results of the regression method. Figure 3B showed the ridge construction as a set of points on each longitudinal axis constructed. The lighter silhouette shows the silhouette of the cow, represented as a cloud of points, disregarding the Z-axis. Figure 3A is an initial ridge line plotting by linear regression.

To determine the cow's height, it is necessary to estimate the coordinates (xyz) of each point along the ridge line and find the extremum along the Z-axis. The point that is the extremum is the withers from which the cow's height is determined.

Table 2 shows the two approaches to BCS evaluation, the upper part was used for BCS evaluation by the expert panel, the lower part of the table was used for automatic evaluation. The numerical values were determined manually by analyzing the resulting field database of animals. The numerical characteristics are the average values for each body condition score and are relevant for the black-motley and the Holstein black-motley breeds raised in Central Russia. For other breeds, the numerical characteristics described in Table 2 may differ [32–34].

**Table 2.** Methods for assessing areas of interest in determining the body condition score of dairy cows by the expert panel and using automated methods.

**Body Condition Rating Table for Experts**

| Current body score condition | Obesity | Above average | Medium | Below average | Exhaustion |
|---|---|---|---|---|---|
| Score | 5 | 4 | 3 | 2 | 1 |
| Spinous processes of the lumbar and back | The back is rounded, hidden in adipose tissue | The back is straight, do not protrude | Raised back, slightly protruding | Visibly protrude, each process is visible | Customized, prong top |
| Transverse process of lumbar and hunger hollow | Hidden in adipose tissue, the area of the fossa is rounded, filled | Smooth rounded edge, the area of the fossa is filled, not sore | Viewed separately, viewed pit | They stand out noticeably, you can count them. The hole is clearly visible | They protrude strongly, the vertebral bodies are visible, the emaciated state, the fossa is deep |
| Hips and pin bone | Hidden in adipose tissue, not visible, ridge is rounded, filled with adipose tissue | Rounded but slightly prominent. Smooth surface | Protrude not sharply, moderately filled | Visibly protruding, thin layer of soft tissue | Protrude strongly |
| Head of tail | Hidden in adipose tissue | Rounded, moderately in adipose tissue | Smooth, covered with soft tissues, adipose tissue is fragmented | Tail vertebrae protrude | Protrude strongly |
| Vulva and anus area | Filled and forms a fat fold | filled | In the form of a small cavity | The cavity is deep, rounded | Protrude strongly, deep depression |

**Body condition score criteria table for automatic scoring**

| Current body score condition | Obesity | | | | Above average | | | | Medium | | | | Below average | | | | Exhaustion |
|---|---|---|---|---|---|---|---|---|---|---|---|---|---|---|---|---|---|
| Score | 5 | 4.75 | 4.5 | 4.25 | 4 | 3.75 | 3.5 | 3.25 | 3 | 2.75 | 2.5 | 2.25 | 2 | 1.75 | 1.5 | 1.25 | 1 |
| h1.Spinous processes of the lumbar and back | not allocated | | | | not allocated | | | | <0.01 m | | | | 0.01 m | 0.013 m | 0.016 m | 0.02 m | 0.02 m< |
| Transverse process of lumbar | not allocated | | | | not allocated | | | | <0.018 m | | | | 0.018 m | 0.021 m | 0.023 m | 0.025 m | <0.025 m |
| Hips and pin bone | not allocated | | | | 145° | 143° | 140° | 138° | 135° | 132° | 129° | 127° | 125° | | 125°> | | |
| h2. Hunger hollow | 0.06 m | | | | 0.07 m | | 0.08 m | | 0.09 m | 0.1 m | 0.11 m | | 0.12 m≥ | | | | |
| h3. Head of tail | 0.05 m | 0.06 m | 0.07 m | 0.08 m | 0.09 m | 0.1 m | 0.11 m | 0.12 m | 0.13 m | 0.15 m | 0.17 m | 0.19 m | 0.2 m≤ | | | | |
| Vulva and anus area | minimum convex | | | | low convex | | | | average convex | | | | maximum convex | | | | |

When the system evaluates the spinous processes of the lumbar and back, and the transverse processes of the lumbar and dorsum, the system first draws straight lines along the ridge and parallel to the ridge lines in the area of the transverse processes of the lumbar then measures the pixel height along the lines (Table 2, side view, node B, parameter h1).

In the hip's area, the system assesses the angle: two lines are drawn along the protruding parts of the back and then the angle is assessed (Figure 4, fatness score 3). The angle at 136° is an indication of a body condition score of 3, and the angle 125°> is a fatness score of 1.75 to 1.

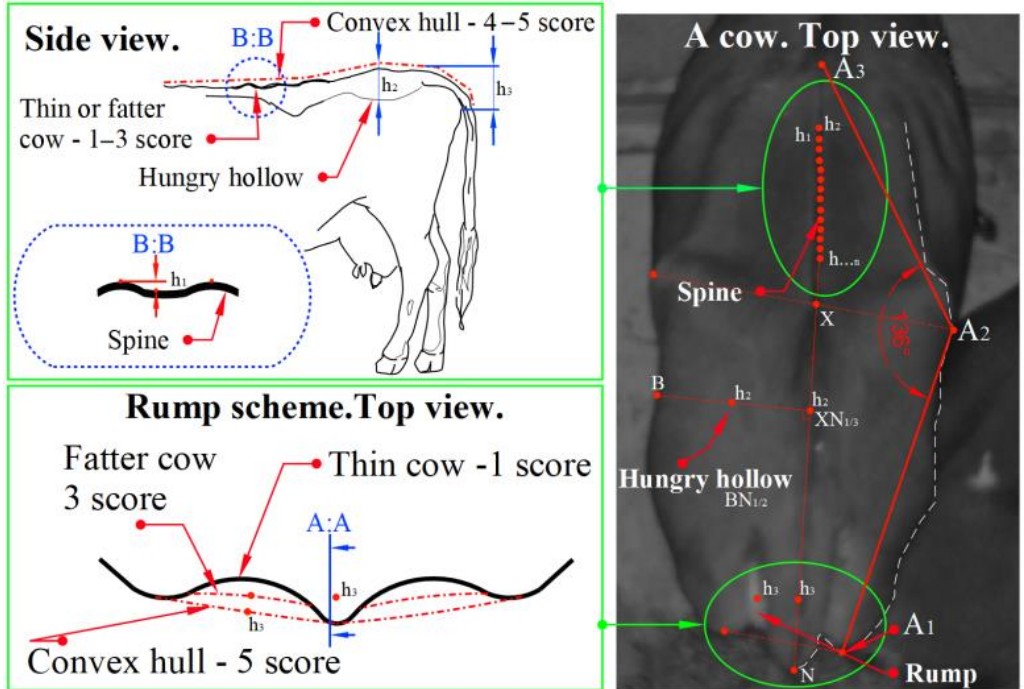

**Figure 4.** Desired areas of interest.

The points for estimating the angle are plotted on the boundary of the protruding parts of the body: the rump bone is $A_1$, the hip protrusion is $A_2$, and the first point at the junction of the transverse processes is $A_3$. To find the point $A_3$, we applied neural network tools with the preliminary training on 80 animals in the 5–6 months of lactation with a body condition score of 1–3 points.

To determine the depth of the "hunger hallow", the following procedure was used: step 1—point XN1/3, which is 1/3 of the length of the segment XN; step 2—at 1/2 the length of the segment BXN1/3, set point h2. Then, we compare the difference in height between points h2. The depth of the "hunger hallow " for a 5-point animal is 0.06 m and for a 1-point animal the depth of the hunger hole is 0.12 m.

The h3 points are determined by the lowest point in the tail base and the highest tail base.

As the last step before determining the fatness, the system checks all criteria and determines the body condition score on a 5-point scale in 0.25-point steps.

Figure 5 shows three-dimensional images converted into the black-and-white format. The pictures show animals with a body condition score from 1 to 5 on a 5-point scale and an explanation of which area of the animal's back is manipulated by the algorithm.

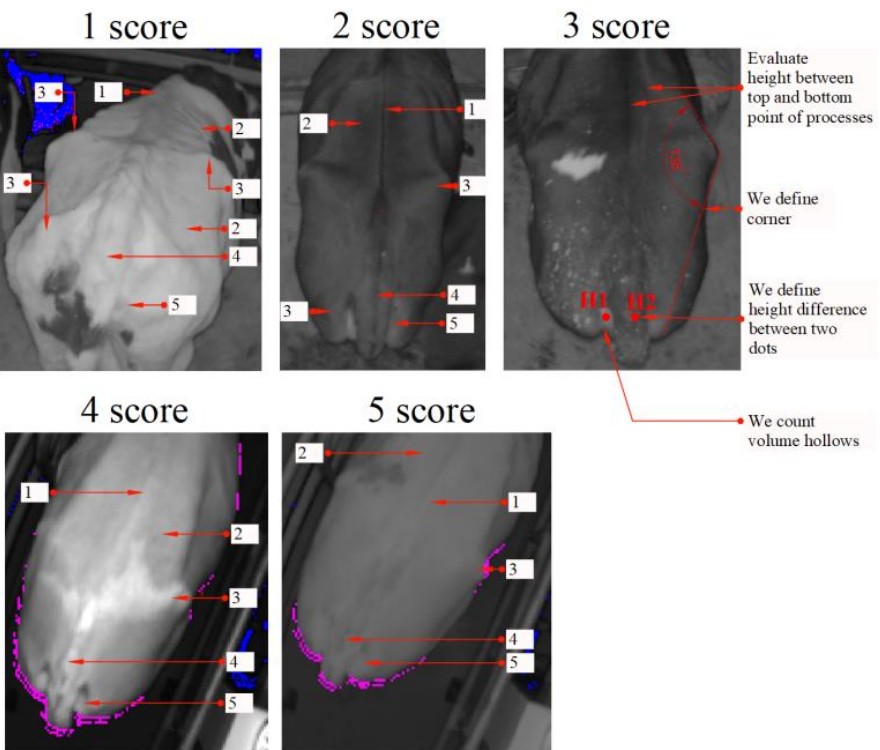

**Figure 5.** Animals' body condition scores and areas of interest. 1—Spinous processes of lumbar and back/dorsum; 2—transverse processes of lumbar and hunger hollow area; 3—hips and pin bone; 4—head of tail; 5—vulva and anus area.

### 3. Results and Discussion

*3.1. Results*

To understand the developed system's overall effectiveness, it was necessary to analyze and evaluate the effectiveness of each area of interest. All the resulting field data were evaluated using the developed method. The results were compared with the experts' evaluation. To understand the overall effectiveness of the developed system, it was necessary to analyze the evaluation effectiveness of each area of interest. To this end, a graph was plotted (Figure 6). The graph shows in the percentage terms the areas of interest and their detection probability, where 0% was not detected in all animals and 100% was detected in all animals.

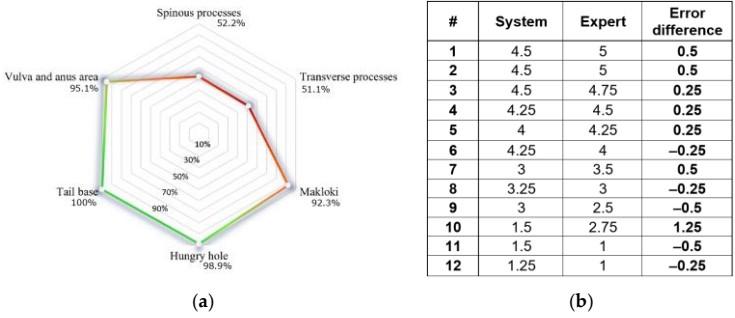

| # | System | Expert | Error difference |
|---|--------|--------|-------------------|
| 1 | 4.5 | 5 | 0.5 |
| 2 | 4.5 | 5 | 0.5 |
| 3 | 4.5 | 4.75 | 0.25 |
| 4 | 4.25 | 4.5 | 0.25 |
| 5 | 4 | 4.25 | 0.25 |
| 6 | 4.25 | 4 | −0.25 |
| 7 | 3 | 3.5 | 0.5 |
| 8 | 3.25 | 3 | −0.25 |
| 9 | 3 | 2.5 | −0.5 |
| 10 | 1.5 | 2.75 | 1.25 |
| 11 | 1.5 | 1 | −0.5 |
| 12 | 1.25 | 1 | −0.25 |

(**a**)  (**b**)

**Figure 6.** Efficiency of the system when detecting the areas of interest in the studied animals. (**a**) measurement efficiency of detecting the areas of interest; (**b**) difference in the BCS evaluation between the automatic measurement of the developed system and the evaluation made by the expert group and the difference between the obtained values.

The graph analysis results of Figure 7 show that the developed method can estimate the tail base area with the 100% accuracy. The hunger hollow is determined with a 98.9%

accuracy and the vulva and anus area with a 95.10% probability. Protruding vertebrae—namely, spinous processes and transverse processes—are evaluated with a 52.20% and a 51.10% accuracy. The accuracy of 50% was explained by the fact that according to Figure 2, these areas were not determined or were determined incorrectly in animals with a body condition score ranging from 3.25 to 5. The overall accuracy of the system was estimated by the experts at 93.4%, which was a positive result.

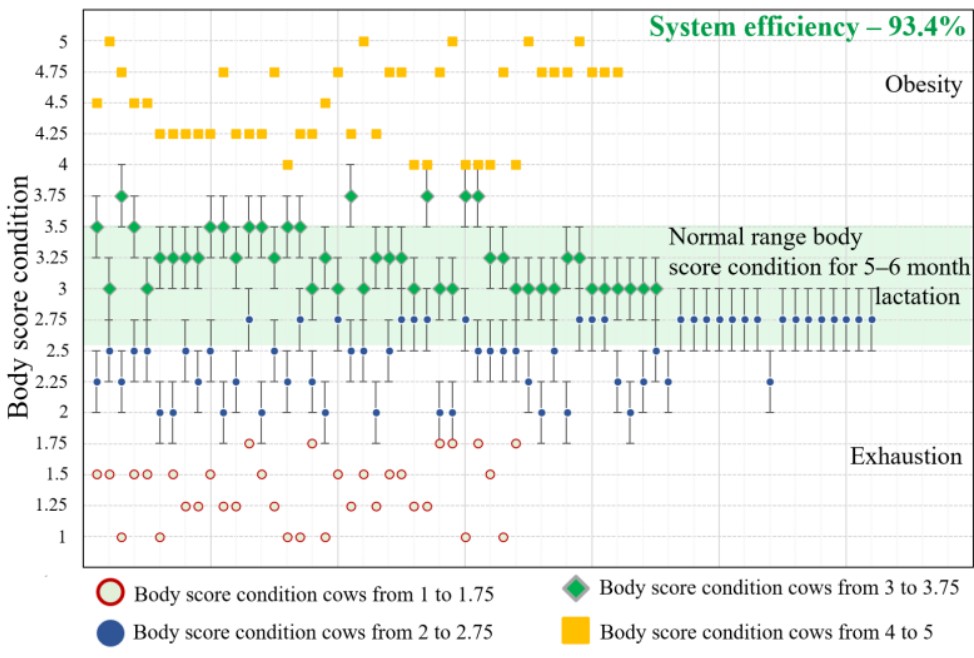

**Figure 7.** Distribution graph of the fatness of 5–6 month old animals obtained in an automatic evaluation.

Additionally, Figure 6 showed that the evaluation of the system and that of the experts have more discrepancies when the body condition score is 4–5, with the largest error of 1.25 and the smallest error of 0.25.

Figure 7 shows that animals with a condition score of 2.5 to 3.5 at 5–6 months are healthy. The developed system gives reasons to divide the animals into three groups, confirming their physiological status: normal range body condition, exhaustion, and obesity. In this case, it is worth bearing in mind that the system has an accuracy of 93.4%. Then, in this study, 4 animals with a body condition score of 3.75, and 15 animals with a body condition score of 2.5 had a 6.6% probability of belonging to another physiological status group. This is due to the fact that the system was wrong by 0.25. Errors caused by other nutritional scores are not critical, as technologically, an animal is either healthy and does not require any manipulation even though the system gave a nutritional score of $3 \pm 0.25$, or it has exhaustion/obesity, which requires manipulation of the animal to improve its physiological status.

In our observations, most of the animals with a fatness score of 3.75–5 were on the second farm (percentage of the total herd when ranked by score) with an average annual milk yield per cow per day of 15 kg/milk and a fat content of 5.1–6%. This farm was financially sound, and the main activity was getting milk from the animals for cheese production. When analyzing the cause of overweight animals, it was found that the farm staff were disrupting the feeding ration and the animals were receiving more energy than they needed; the animals were kept in loose housing.

On the second farm, a correlation was established between live weight and body condition score for 32 animals (Figure 8). The correlation determined by Pearson's method is R = 0.849.

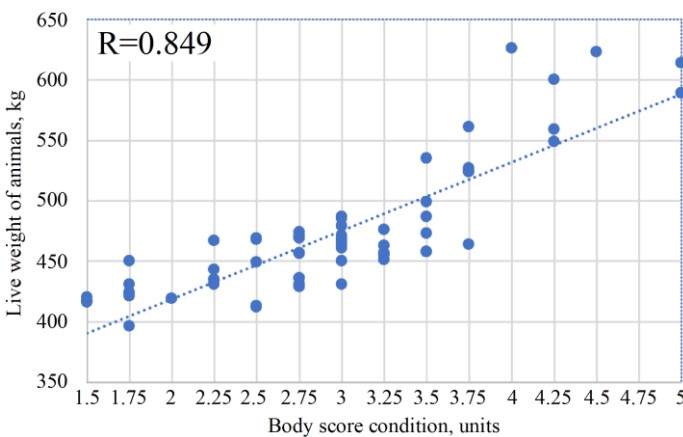

**Figure 8.** Distribution graph of the fatness of 5–6-month-old animals obtained during an automatic evaluation in 32 animals.

The Pearson correlation tool was chosen to determine if there was a relationship between the live weight and body condition score of the cows under study, as the data obtained have a normal distribution. Discussion of the data shows that part of the values on the scale from 4 BCS points to 5 BCS points and a live weight above 525 kg have a chaotic distribution. The Pearson correlation is R = 0.849, which does not guarantee 100% correlation. This is explained by the following: the live weight of animals consists of basic parameters—the amount of dirt accumulated on the animal, the amount of feed eaten, and the month of pregnancy. Additionally, live weight was obtained using the Klüver–Strauch method [33], where the method itself has a margin of error. This correlation did not allow the evaluation of live weight by the fatness score, but is only an additional signaling indicator that draws attention to live weight gain. Therefore, by analyzing the data by the Pearson correlation, our main aim was to understand if there is a relationship between obesity and weight gain. This was important for the purpose of additional animal monitoring, where the developed software will signal if an animal is overweight, which in turn negatively affects the probability of successful insemination. If it was detected that an animal is gaining excessive live weight, it was therefore necessary to move the animal to another housing group to change the feeding ration.

Studies [33] found that an optimal range of body weight for an increased performance does exist due to the non-linear relationship between milk yield and body weight. Dairy breeds respond more strongly to bodyweight range than dual-purpose breeds. Cows with an average weight are the most productive in the population. Heavy cows (>750 kg) produce much less milk. Special attention should, therefore, be paid to the daily ration, and further increases in body weight of dairy cows should be avoided. Animals with a body condition score of 1 to 2.5, in most cases, were found on a farm with an average annual milk yield per cow per day of 16.8 kg/milk, and a fat content of 3.6 to 3.8%. After examining the keeping conditions of the animals, several criteria influencing the emaciation of the animals were observed. The main criterion was the feed ration. The animals under study received mainly legume–grass hay with the addition of micro and macro nutrients in their diets. The animals were continuously fed a complete daily ration consisting of 4 kg of legume–grass hay, 15 kg of mixed grass silage, 6 kg of root crops, 5 kg of high energy mixed fodder, 1 kg of barley powder, and 50 g of table salt. In addition, it was recorded that animals were kept in concrete buildings, typical for buildings constructed in the 1980s, with a disturbed microclimate and tethered housing without regular walks.

Based on the research results, the algorithm for the automatic evaluation of the animal body condition (fatness), followed by the staging of their physiological condition, was supplemented and modified. The algorithm was divided into two parts and is shown in Figures 9 and 10. The second part of the algorithm is an integral part of the first one. The algorithm was included in the software code of the developed software.

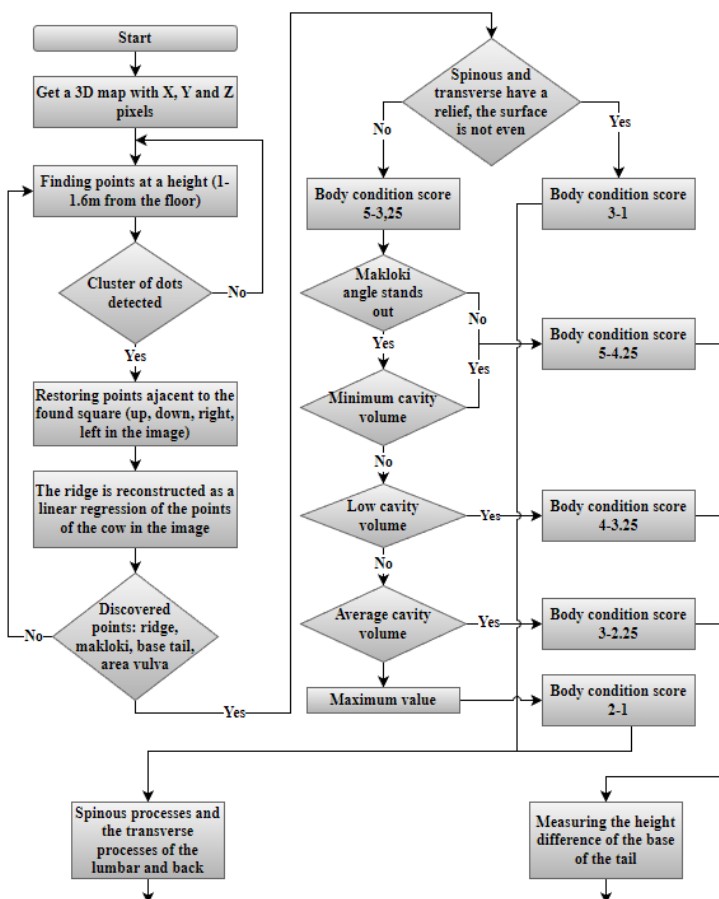

**Figure 9.** First part of the BCS Algorithm.

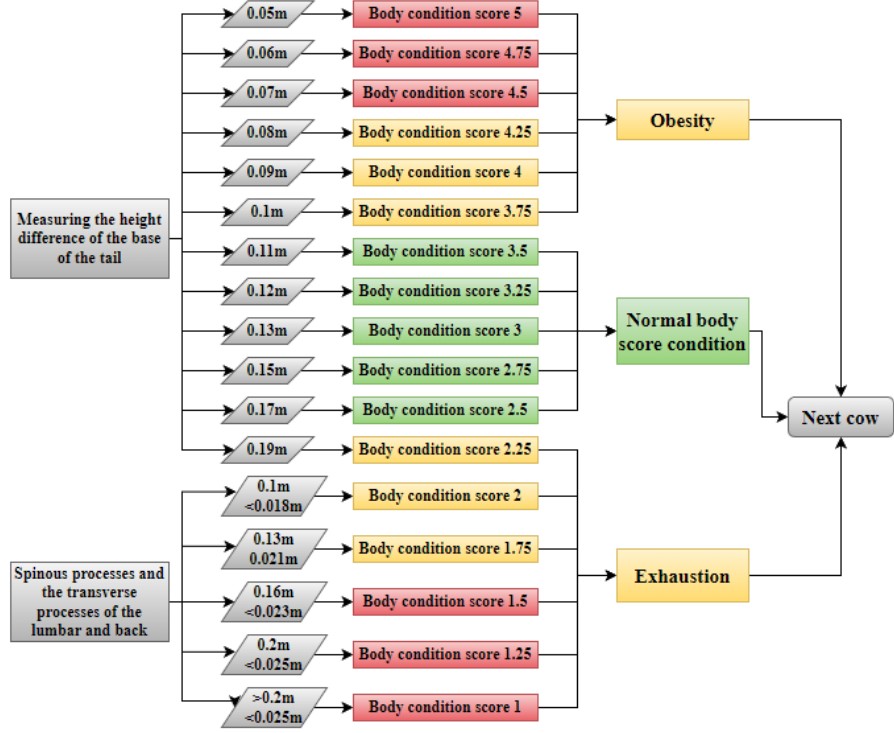

**Figure 10.** Second part of the BCS algorithm.

The explanation of Figure 9 starts with a three-dimensional map containing X, Y and Z coordinates of each pixel, and then searches for points at height (in the range of 1–1.6 m from the floor). When a cluster of points is detected, the algorithm determines the location of the main features: the spine, the hips, the tail head, and the vulva region. The topography of the spinous and transverse processes relief was then determined and the initial BCS value was determined.

As for Figure 10, for the initial BCS of 2.25–5 with no scalloping of the spinous and transverse processors and a low depression, the difference in tail head height was measured. For the primary BCS values in the range 1–3 with spinous and transverse processes in relief and a large hollow volume, the spinous and transverse processes of the lumbar and dorsum were measured. Depending on the results, the algorithm outputs had three evaluation options—'normal BCS', 'obesity', 'exhaustion'.

The developed software gave information about the animal by scanning RFID tags: date and time of the last measurement, sex, status, date of birth, actual weight, weight excluding animal contamination (bulk), weight excluding GIT contents, animal height, and the BCS value. The software developed for each animal provided more detailed information, traced the dynamics of changes in physical parameters, kept the herd log, and had service settings.

*3.2. Research limitation*

We refer to several factors as limitations of the research.

Factor 1 is the capability of the machinery and equipment and the environmental conditions in which they were operated. Dairy cows were evaluated both indoors and in the open field. Based on our experience with the equipment, we have found that 3D TOF cameras with a 840 nm wavelength, when shooting animals outdoors in bright sunlight, had noise that prevented effective fatness scoring. As such, 3D TOF cameras at 940 nm may be considered for further research. According to the manufacturers (the study does not specify a specific manufacturer), the 940 nm 3D cameras solve the problem of not being able to produce 3D maps in bright sunlight. In this study, three-dimensional cameras based on 940 nm were not tested.

Factor 2—while evaluating the body condition score of an animal, we could not estimate the animal's weight to an accuracy of 1 kg. We considered it possible to install an additional 3D camera to measure the torso depth of the animal—automatically using the Klüver–Strauch method [32] based on the digital data obtained for the torso depth, height and body condition score. However, this method would also not give accurate information about the animal's live weight, as there was no information about the cow's pregnancy, degree of contamination, and gastrointestinal contents. Additional discounts and coefficients relative to live weight would have to be introduced, but this may result in a high margin of error.

Factor 3 is the use of artificial intelligence to find the areas of interest. The matter is that if all fields of interest are calculated by means of artificial intelligence, then the exploitation of the system with each new breed or farm will demand resources for additional training of the system, which is not practical. Using the proposed research method for code development is a more labor-intensive process than collecting a data array and training artificial intellect. However, this method would definitely prove to be more practical, because it covered dairy cattle breeds, which are bred in Russia.

Factor 4 is the use of specific equipment. For these studies, it was not the specific manufacturer of the 3D cameras that was important, but their characteristics. It was important to choose a 3D camera that has a wavelength of 840 nm and a resolution of $352 \times 264$, and the factory error rate of used cameras is not higher than 1 cm per 1 m distance at a distance from the object in question.

### 3.3. Economic Efficiency

The proposed technology will improve production efficiency on large dairy farms by reducing animal stress, controlling animal nutrition when necessary, and early detecting physical deviations (Table 3).

**Table 3.** Cost of implementing the technology using the example of the farms under study.

| Criteria | Used Solutions | | | Proposed Technology | | |
|---|---|---|---|---|---|---|
| | **1st Farm** | **2nd Farm** | **3rd Farm** | **1st Farm** | **2nd Farm** | **3rd Farm** |
| The number of cows, heads | 560 | 50 | 1200 | 560 | 50 | 1200 |
| Milk yield, kg | 16.8 | 15 | 28 | 40–50 | 40–50 | 40–50 |
| Culling, % | 7 | 7.5 | 6 | 4 | 4 | 4 |
| Die, % | 5.5 | 6 | 4 | 1.2 | 1.2 | 1.2 |
| Feed consumption, t/day | 16.8 | 3.3 | 54 | 28.2 | 2.5 | 60.4 |
| Veterinary care costs, rub/month | 157,300 | 78,650 | 235,950 | 18,000 | 12,000 | 25,000 |
| Veterinary care costs, rub/year | 1,887,600 | 943,800 | 2,831,400 | The system installation's price | | |
| | | | | 2,515,968 | 2,515,968 | 2,515,968 |
| Number of calves, head | 333 | 29 | 714 | 448 | 40 | 960 |
| Calves for sale (80%), heads | 266 | 23 | 571 | 358 | 32 | 768 |
| Calves for sale (1 month, 60 kg), profit, rub | 2,397,600 | 208,800 | 5,140,800 | 3,225,600 | 288,000 | 6,912,000 |
| Calves for sale (6 months, 140 kg), profit, rub | 3,916,080 | 341,040 | 8,396,640 | 5,268,480 | 470,400 | 11,289,600 |
| Calves for sale (12 months, 350 kg), profit, rub | 4,195,800 | 365,400 | 8,996,400 | 5,644,800 | 504,000 | 12,096,000 |
| Total profit, rub | 8,358,180 | −116,710 | 19,238,490 | 11,565,712 | −1,269,068 | 27,672,632 |

The percentage of culling and mortality was planned to be reduced by adjusting the ration and improving the general maintenance condition of the animals on the farms. We also proposed to increase the actual milk yield per day.

Often, farms have in-house veterinarians, but with the introduction of the biometric system, costs can be reduced, and external specialists can be called in only when necessary. Feed costs would also be reduced, as feed rations for the animals can be monitored and adjusted.

The main profit increase was expected to come from the improved life quality of the cows, and as a consequence, the birth rate of calves will also increase.

Sales are planned by age groups. The distribution will be as follows: 80% of all calves born on the farm during the year will be sold. Of these, 50% will be sold at the age of 1 month, 35%—6 months, and 15%—12 months.

As far as in the first year, Farm 1 and Farm 3 would make 38.4% and 43.8% more profit, respectively, but this technology did not look profitable on the second farm. We recommend that this biometric system should only be installed on large farms with 560 heads or more.

### 3.4. Technology Applicability

Having confirmed the cost-effectiveness of the developed BCS estimation technology, we can now describe how we implement automatic BCS evaluation for milk production.

The automatic livestock monitoring system operated in two ways. The first way was stationary, and the second way was mobile.

The stationary method consisted in the fact that on the farm, in the places of the daily pass of the animals, for example, the system of BCS evaluation was mounted behind the

milking parlor in the "gallery". The system consisted of a three-dimensional camera and data collection and processing unit, as well as an identification antenna, which read the ID number of the cow. The data were sent to a server.

The mobile method implied bringing the system once a month to the box where a group of animals was kept to scan the BCS score (Figure 11).

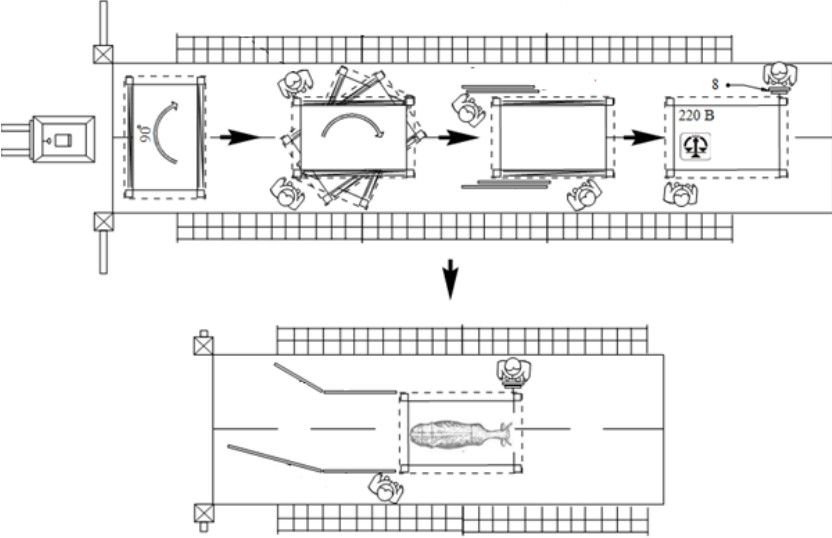

**Figure 11.** Schematic diagram of the installation of a mobile BCS evaluation system in a cubicle housing a group of animals.

The system was brought in by a forklift or an ATV to the cubicle where the animals were kept in a loose housing. Two staff members then turn the system around; they set the fence in the desired position, pointing to one side or the other. Then, one employee drove the animals in and out of the system, a second employee took care of reading the cow number, assessed the body condition, and recorded the data. When the group was finished, the system was assembled in the transport position and transported to the next group of animals. The data were transferred on a flash drive to a server. This procedure is done on a monthly basis. The advantage of the mobile system is that the fatness estimation can be done while the animals are grazing in the fields.

On the basis of the data obtained, the developed software plots a graph—a diagram of the change in the BCS score—and compares it with the set-required values for the current physiological status of each cow. There were several applications of the technology. The first situation was when we had an animal with an increased body condition score. The system recorded that the BSC score was increased, then queried the following data from the herd management software: current physiological status, which group the animal is in, current milk production, day of lactation, insemination status, and specific ration. For example, an animal was on day 75 of lactation, no conception had occurred, the BCS score was 3.75, milk yield was 17 kg/day, and fatness was 3.7%. Then, an automatic decision was made that the ration should be adjusted by reducing the amount of energy the animal receives without changing the animal's maintenance group, as the animal was at the peak of lactation and its milking requirements should be met. When moving to the next group, a gradual decrease in milking should be observed, accordingly. At the same time, we have to monitor the animal's condition so that by the end of lactation, the animal has a corrected condition. For example, an animal on day 190 of lactation and the conception on day 110, the BCS condition score was 2.5, milk yield 14 kg/day, and fat content 3.5%. In this case, the animal should be moved from group 3 to group 1 or 2 in order to adjust the feeding level to ensure an energy surplus.

The BCS evaluation system was needed as an additional tool to monitor feeding and assist in decision making for each cow when moving them to different housing groups.

The development of an automatic fatness estimation system will make it possible to collect data sets and statistics for each animal. This will make it possible, when collecting data on feeding, animal genetics, breeding material, and diseases, to form animal groups on farms more effectively, revealing their genetic potential in terms of productivity.

## 4. Conclusions

The effect of increasing the production of dairy farms had been achieved by implementing the technology of an automatic evaluation of the fatness of dairy herds (BCS) in a 0.25 step on a 5-point scale. The developed technology had been tested on 3 farms, with a total herd of 1810 animals, and provided for a non-contact BCS evaluation of a dairy herd required throughout the life of the herd within the farm. The overall accuracy of the system was estimated at 93.4%. The study has demonstrated the economic effect of implementing the proposed system.

**Author Contributions:** S.S.Y.—project management, methodology, natural data collection, writing—editing. D.Y.P.—conceptualization, software development. I.M.D.—system development, writing—analysis and editing. V.A.P.—visualization, rude preparation. A.A.S.—software testing. Y.A.P.—natural data collecting, editing. I.Y.—natural data collecting, writing—editing. All authors have read and agreed to the published version of the manuscript.

**Institutional Review Board Statement:** Not applicable.

**Informed Consent Statement:** Not applicable.

**Conflicts of Interest:** The authors declare no conflict of interest. The funders had no role in the design of the study; in the collection, analyses, or interpretation of data; in the writing of the manuscript, or in the decision to publish the results.

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
