# Peer review of "Technology of Automatic Evaluation of Dairy Herd Fatness"

_agriculture, doi:10.3390/agriculture13071363_

Round 1

Reviewer 1 Report

The study focuses on the introduction of automatic assessment technology for dairy herd fatness, aiming to improve milk production efficiency on dairy farms. The use of a 3D ToF camera and developed software allows for non-contact evaluation of various body areas of dairy cows. The results indicate high accuracy in assessing certain areas, such as the tail head, hungry pit, and vulva and anus area. However, the evaluation of vertebrae protrusion shows relatively lower probability. Overall, the developed system demonstrates promising potential in categorizing animals based on their physiological status and can contribute to increased dairy farm efficiency.

Abstract: delate the on brackets words

keywords must be revised: artificial is not pertinent, and no abbreviations

line 49-51: i suggest report a reference, i suggest: 10.3389/fvets.2023.1141286

line 53-54: i suggest report a reference, i suggest: 10.1080/1828051X.2022.2032850

line 57 and 60; instead of report in this study report the name of the first author

lines 61-64: please rewrite, is not clear

I would like to suggest a revision of the introduction section of your paper, focusing on the following points:

  1. Importance of body condition score (BCS) evaluation: Start by highlighting the significance of accurately assessing the body condition of dairy cows in order to optimize their health, welfare, and milk production. Discuss how BCS serves as a valuable indicator of nutritional status, reproductive performance, and overall well-being.

  2. Existing knowledge on BCS and its effects: Provide a concise overview of previous research that has investigated the relationship between BCS and various aspects of dairy cow management and productivity. Discuss findings related to milk yield, fertility, disease incidence, and longevity, highlighting both positive and negative effects of different BCS levels.

  3. Aim of the study: Clearly state the specific objective of your research. This could be to evaluate the effectiveness of the developed automatic assessment technology for dairy herd fatness, with a focus on its accuracy and potential impact on milk production efficiency. Briefly mention the methodology employed, such as the use of a 3D ToF camera and developed software, and the data collection process.

specify the breed brown Latvian

do not specify black and white Holstein, everyone knows the color of Holsteins cows

report the reference for the fatness score and state the difference with the BCS

3 farms, commercial farms?

avoid the private names in the methods, put them of acknowledgment

report the breed of the animals in the farms

report DIM

report average fat and not range

report feeding conditions

report average and SD BW and not range

report the reference for the method on line 162

In addition to the suggestions provided earlier, I would like to recommend including a section on practical implications and limitations of your study. This will help contextualize your findings and provide valuable insights for researchers and practitioners in the field. Here's how you can approach this:

Practical implications: Discuss the potential practical applications of your developed automatic assessment technology for dairy herd fatness. Highlight how this technology can be integrated into on-farm management practices to enhance the efficiency of milk production. Emphasize the benefits of non-contact evaluation and the ability to assess multiple body areas accurately. Additionally, consider discussing how the system can aid in identifying and managing animals in different physiological states, such as normal, exhausted, or obese.

Limitations of the study: Acknowledge any limitations or constraints that might affect the interpretation of your findings. This could include limitations related to the sample size, data collection period, or specific characteristics of the farms involved. Discuss any potential sources of error or variability in the assessment process and how they may have influenced the results. It is important to provide a balanced view of the study's limitations to ensure transparency and accurate interpretation of the findings.

By including a section on practical implications and limitations, you will provide a more comprehensive discussion of your study's relevance and potential application in real-world dairy farming settings, while also acknowledging any constraints or areas that require further investigation.

Author Response

Thank you very much for appreciating the importance of our paper and good comments. Below we reply to all points.

Reviewer 2 Report

A 10% of the population of cows with 5 to 6 months in lactation was selected, which amounts to 182 animals from three populations. Live weight was related to the fat score (However, it is not clear if  the model's assumptions were met). Based on photographs, the herd's fatness score was obtained (important to mention that this variable is measured on a weak scale but treated as a strong scale).

Based on the latter comments, I have two main questions concerning the methodology and results:

1.       Doubts arise when talking about cause-effect relationship based on Pearson correlations, that is imprecise because the correlation between variables does not determine cause-effect relationships but rather the degree of association.

2.       While the evaluated variable (fat score) is not measured in a strong scale, the authors never mention or consider using free-distribution statistics. Furthermore, testing the model assumptions should be addressed in the discussion.

Author Response

Thank you very much for the fair comments. Below we reply to all points.

A 10% of the population of cows with 5 to 6 months in lactation was selected, which amounts to 182 animals from three populations. Live weight was related to the fat score (However, it is not clear if  the model's assumptions were met). Based on photographs, the herd's fatness score was obtained (important to mention that this variable is measured on a weak scale but treated as a strong scale).

Reviewer 3 Report

The study aims to forecast the dairy farm producing efficiency through automatic evaluating technology of dairy herd’s fatness at a proposed point scale.

The idea is quite interesting, although not a total novelty. The primary issues of this paper are: 1) it is poorly written with several language mistakes, and 2) no Discussion is provided comparing the current study results with previous studies.

My decision is for a significant review. The authors should provide a serious review of the paper by an academic with experience in scientific writing. The Results and Discussion session should compare and provide a scientific discussion with the present study and those previously published and mentioned in the Introduction.

Please check the attached file for the corrections. 

The manuscript lacks scientific language and it needs improvement.

Author Response

Thank you very much for your positive evaluation of our work and good comments. Below we reply to all points.

The study aims to forecast the dairy farm producing efficiency through automatic evaluating technology of dairy herd’s fatness at a proposed point scale.

The idea is quite interesting, although not a total novelty. The primary issues of this paper are: 1) it is poorly written with several language mistakes, and 2) no Discussion is provided comparing the current study results with previous studies.

Round 2

Reviewer 1 Report

after the revision the paper improved a lot, I endorse the pubblication

Author Response

Thank you for your positive evaluation of our study. 

Reviewer 3 Report

The theme is interesting, and it represents some degree of innovation. I understand it is sound scientific research, however, scientific writing is still inadequate and requires a serious review.

My recommendation is for major revision yet.

The scientific writing is inadequate and requires a serious review. For instance, a comma is used instead of a period when referring to values in numbers. 

The are several short paragraphs in the introduction that should merge. When referencing previous studies, citing references in the text is weird and does not reflect proper scientific language. I recommend a profound review of the language.

My recommendation is for major revision yet.

Author Response

We have corrected the English throughout the text. We replaced the «comma» separator with a «dot» separator. We look forward to your positive decision on our study. Thank you for your work and your time.
